# In Vitro and In Vivo Evaluation of the Effects of a Compound Based on Plants, Yeast and Trace Elements on the Ruminal Function of Dairy Cows

**Francoise Lessire** [1,*] 🆔, **Sandra Point** [2], **Anca-Lucia Laza Knoerr** [2] and **Isabelle Dufrasne** [1,3,*]

[1] Centre des Technologies Agronomiques, 4577 Strée, Belgium
[2] Centre Mondial de l'Innovation–AII (Agro-Innovation-International), Roullier, 35400 Saint Malo, France; sandra.point@roullier.com (S.P.); ancalucia.lazaknoerr@roullier.com (A.-L.L.K.)
[3] FARAH Department of Veterinary Management of Animal Resources, Faculty of Veterinary Medicine, University of Liege, 4000 Liège, Belgium
[*] Correspondence: flessire@uliege.be (F.L.); isabelle.dufrasne@uliege.be (I.D.)

**Abstract:** The high production levels reached by the dairy sector need adjustment in nutritional inputs and efficient feed conversion. In this context, we evaluated a compound (QY—Qualix Yellow) combining optimized inputs in trace elements and 20% MIX 3.0. In a first step, the effects of MIX 3.0 on ruminal function were assessed in vitro by incubating ruminal fluid with the mixture at a ratio of 20:1. The results obtained encouraged us to test QY in vivo, on a herd of dairy cows. The herd was divided into one group of 19 dairy cows receiving the compound and a control group of 20 animals conducted in the same conditions, but which did not received the compound; the production performance and feed efficiency of the two groups were compared. In vitro experiments showed improved digestion of acid and neutral detergent fibres by 10%. The propionate production was enhanced by 14.5% after 6 h incubation with MIX 3.0. The plant mixture decreased the production of methane and ammonia by 37% and 52%, respectively, and reduced the number of protozoa by 50%. An increase in milk yield by 2.4 kg/cow/d ($p < 0.1$), combined with a decrease in concentrate consumption of 0.27 kg DM/cow/d ($p < 0.001$), was observed in vivo after consumption of the compound. Sixty-six days after the beginning of the trial, methane emissions per kg of milk were significantly lower in the group receiving QY. In conclusion, MIX 3.0 induced change in ruminal function in vitro and, when it entered into the composition of the QY, it appeared to improve feed efficiency and production performance in vivo.

**Keywords:** methane; dairy cows; trace elements; ruminal function; feed additive

## 1. Introduction

Worldwide, we can observe a rise in the milk yield (MY) of dairy cows [1,2], which is increasing to meet nutritional needs. Conversely, the agricultural and, particularly, the livestock sector are regularly targeted because of their contribution to global anthropic greenhouse gas (GHG) emissions [3]. The contribution of agriculture is estimated at around 14.5%, with amounts varying depending on the intensification of the livestock sector. Emissions composed mainly of enteric methane and nitrogen (N) excreted by livestock contribute to global emissions of greenhouse gases and ammonia. Mitigating methane emissions and reducing nitrogen losses are therefore major concerns from an environmental as well as an economic point of view [4,5]. Optimizing ruminal function is thus of great interest for minimising methane emissions and maximising the production performance of dairy cows [6]. Moreover, high production levels also require adjusted nutritional inputs in minerals and trace elements, e.g., iodine (I), zinc (Zn), copper (Cu), selenium (Se), cobalt (Co) and manganese (Mn) [7,8]. Indeed, these elements are included in the structure of multiple enzymes and proteins so that a deficiency can lead to a broad range

of health disorders, including anaemia, poor reproductive performance and low immunity. Most of these play a role in the prevention of oxidative stress [9,10]. Thus, the nutritional requirements of trace elements have been re-evaluated on the basis of higher nutritional demand, coupled with enhanced production [11] so that supplementation is in most cases necessary for high-producing cows.

MIX 3.0 is a formulation developed by Roullier which is included in a 20% mix in the commercial product QY and is composed of a mixture of yeast, plant extracts and aromatic compounds. The plant extracts include thyme (*Thymus vulgaris* L.), garlic (*Allium sativum* L.), absynthe (*Arthemisia absynthum* L.), male fern (*Dryopteris A.* spp.), goosefoot (*Chenopodium quinoa* W.), tansy (*Tanacetum vulgare* L.), elecampane (*Inula helenium* L.) and boldo (*Peumus boldus* M.). Qualix Yellow is marketed as a lick bucket whose composition includes 20% MIX 3.0 and various oligo and macro elements i.e., Zn, Mn, Cu, Co, Se, and in I. The objectives of this study were, in a first step, to evaluate in vitro the effects of the MIX 3.0 on the ruminal fermentation pattern. In a second step, we tested the effects of Qualix Yellow (QY). According to the results obtained in vitro with the MIX 3.0, supplementation with QY is expected to modify ruminal fermentations, to increase feed conversion efficiency and to adjust trace element inputs to the nutritional needs of grazing dairy cows. To test this hypothesis, we compared the production parameters and methane emissions of a group of dairy cows with access to the compound to those of a control group.

## 2. Materials and Methods

### 2.1. In Vitro Trials

The in vitro study was carried out at the International Centre of Research of Roullier Groupe (Saint Malo, France) in August 2016. Ruminal fluid was collected on 5 fistulated cows from the experimental farm of Méjusseaume—INRAE (Rennes, France, (1.47° W 48.70° N). The animals received a ration composed of maize silage (50%), concentrates (10%) and meadow hay (40%) on a dry matter basis twice a day and had free access to water and mineral blocks. Ruminal fluid was collected before the morning meal and filtered through two metal sieves (1 and 0.4-mm mesh). The samples of all the animals were mixed and kept under anaerobic conditions at 39 °C until further analysis. The sample was then buffered with artificial saliva in a proportion of 1:2, according to the protocol described by Menke and Steingass [12] and incubated in anaerobic conditions at 39 °C with 0.5 g DM of cows' diet composed of 50% maize silage, 30% meadow hay and 20% concentrate rich in energy on a dry matter basis, then 24 mg of MIX 3.0, provided by Roullier Groupe, were added. The final volume in the flask was 60 mL, so that the final concentration of MIX 3.0 was 0.4 mg/mL of inoculum. Four flasks of control + 4 flasks containing inoculum were incubated. The incubations of inocula were repeated 4 times. Gas production was measured by the method described by Menke et al., 1979 [12,13] and developed by the laboratory Roullier with an Ankom RF gas production system (AnkomTechnology, Macedon, NY, USA), used in accordance with the Ankom. Technology Instrument and Procedure Manuals (2010).Production of methane and $NH_3$ and the counting of protozoa were performed 6 h and 24 h after the incubation start. The concentration of methane was measured by micro-GCMS (Agilent 1260, Agilent Technologies, Ltd., Santa Clara, CA, USA) in the laboratory OSU (Rennes, France). After sampling from different flasks through filter paper, ammonia was determined by titration after distillation with a Buchi SpeedDigester K439 (Büchi AG, Flawil, Switzerland), used for the determination of proteins according to AOAC [14], 6 h and 24 h after the start of fermentation. The counting of protozoa was made after sample fixation with formaldehyde (18%) and by reading on a Malassez cell by microscopy (×10) and was repeated 3 times. Fermentations were stopped by freezing the samples. Analysis of VFA was performed in the laboratory UPsciences (Saint Nolff, France). They were measured by CPG with HPFFAP column and FID detector (Agilent 1260, Agilent Technologies, Ltd., Santa Clara, CA, USA). Fibre digestibility (acid detergent fibre (ADF) and neutral detergent fibre (NDF)) was quantified by measuring NDF and ADF by Van

Soest method [15] with a fibre sac (AnkomTechnology, Macedon, NY, USA) at time 0 and 24 h of incubation.

*2.2. In Vivo Trials*

2.2.1. Study Site

The study was carried out at the Centre of Agronomic Technologies (5.31° E 50.507° N) located in Strée (Belgium) from 1 August 2017 to 6 October 2017 for a period of 66 days. An adaptation period of 15 days (17–31 July 2017) preceded the start of the trials. The in vivo experiment was conducted according to Belgian animal welfare rules.

2.2.2. Experimental Design and Animals

Thirty-nine cows were randomly assigned to 2 groups, balanced on the basis of milk yield (MY), recorded over the previous days, days in milk (DIM) and lactation number (LN). The group GQY was composed of 19 cows (DIM: 169 ± 77 days; LN: 2 ± 1 including 7 primiparous; MY: 25.7 ± 4.3 kg/cow/d), while 20 cows (DIM: 164 ± 77 days; LN: 2 ± 1 including 8 primiparous; MY: 26.9 ± 5.4 kg/cow/d) were included in group GC. The groups were physically separated. They grazed different paddocks, and the layout of the barn allowed specific access for each group. The barn was divided into two parts, with an automatic concentrate supplier (ACS) and a row of fences for each group. When the animals returned to the barn for morning and evening milking, they were blocked in their assigned area. Thus, dry matter intake (DMI) of forages and concentrate were measured for each group. A total of 4 buckets containing QY (2 indoors, 2 outdoors) were made available for GQY. Every 3 days, the buckets were weighted to estimate the daily consumption of the cows submitted to the treatment. In summary, the cows of both groups received cereal crop silage (mixture of oats, triticale and peas; DM: 320 g/kg DM; CP: 102 g/kg DM; cellulose: 297 g/kg DM; NDF:519 g/kgDM) and concentrate (DM: 881 g/kg DM; CP: 239 g/kg DM; starch: 364 g/kg DM; sugars: 52 g/kg DM; NDF: 332 g/kg DM) provided at ACS in complementation of grazed grass. The amount of supplied silage was recorded on the mixer feeder wagon, as were refusals. The concentrate supply was adjusted to the recorded MY of each animal, and its consumption was recorded in the ACS. The daily dry matter intake (DMI) at the barn was thus estimated for each group. Sward height was measured on each paddock on a weekly basis when cows came out and in, using an electronic connected rising plate meter (EC20®, Feilding, New Zealand). This method allowed calculation of grass consumption by multiplying the difference in grass height by the sward density (kg DM/cm/ha) and by the area of the paddock. The weekly grass height measurements allowed estimation of the grass growth in order to take this parameter into account in the former estimation. The obtained value was then divided by the number of cows grazing on the paddock. The grass intake values obtained were compared with the nutritional intake calculations to check their reliability. The mineral content of grazed grass and cereal crop silage (oats, triticale, peas) were obtained, after calcination at 450 °C and mineralisation with $HNO_3$ by ICP-OES (inductively coupled plasma optical-emission spectrometer) [16] The allocated diet met the nutritional requirements related to the recorded milk yield in accordance with NRC recommendations [11]. The production performance of each cow in both groups was collected: the daily MY was obtained from DeLaval Alpro® general milking management (DeLaval AG, Sursee, Switzerland) during the duration of the trial. Once a month, milk samples were collected from each milking and sent to the dairy herd controlling system (Association Wallonne de l'Elevage) to determine milk composition (% fat (F), % protein (P), urea (mg/L).

Two methods were performed to assess the methane emissions of the 2 groups. The first one is based on breath samplings. One infrared methane analyser (Guardian Plus; Edinburgh Instruments Ltd., Livingston, UK) was installed in each ACS. Breath samples were collected every 3 s while the cows were eating. Methane production was estimated following the method described by Garnsworthy et al., 2012 [17,18] and by Haque et al., 2017 [19]. The measurements were performed from 19 to 25 September (7 days). They

were assessed in each group in the same automatic concentrate supplier, so it was possible to compare their emissions in the same ambiance conditions. Moreover, devices were changed from one ACS to the other to check that there was no difference attributable to the used apparatus.

The second method is based on methane predictions in milk samples following the methodology described hereafter. Individual milk samples, from morning and evening milkings, were sent once a month (9 August 2017, 6 September 2017, 5 October 2017) to the Comité du lait (a certified milk control station, Battice, Belgium; Belgian accreditation number262-TEST in compliance with ISO 17025) for FT-IR spectral analyses using a MilkoScan FT6000 spectrophotometer (Foss, Hillerød, Denmark). The predictions of emitted methane were performed using the equation developed on milk Fourier transform MIR spectra by Vanlierde et al., 2016 [20]. This equation was validated using data from respiratory chambers [20,21], was regularly updated and its limits, defined in several publications [22,23], were strictly observed in this paper. The results were reported per cow (methane (g)/cow/d) and kg of milk and/kg energy corrected milk (ECM) produced by each animal.

Cows were weighted once a month, and their body condition score (BCS) was noted following the method described by Edmonson et al., 1989 [24]. Every event relative to health condition was reported. Digestive efficiency of the diet was evaluated on faecal samples using the sieving kit Deltavit® (Janzé, France), following the procedure described by Carta, 2010 [25]. The kit was composed of 1 pan and 2 sieves whose meshes were 5 and 2 mm. Faecal grabs were collected at the same time in the morning, from 5 fresh calved cows selected in each group, mixed and then placed in the first sieve. The samples were rinsed with running water until only coarse and medium-size particles remained in the first and second sieve, respectively. The fractions from the different sieves and from the pan were collected and weighed. Values were compared to the literature and between the groups.

### 2.2.3. Pasture Layout

Cows from the 2 groups had access to pasture. The total pasture area was 18.09 ha divided into 8 paddocks: 7 paddocks of permanent grassland with area ranging from 1.59 to 2.02 ha, composed mainly of meadow grass (*Poa trivialis* L.), white clover (*Trifolium repens* L.) and perennial ryegrass (*Lolium perenne* L.), and a paddock of 5 ha of temporary grassland in a rotary cycle of 3 years. Pastures were managed following rotational grazing. Sward density was assessed by mowing a grass band 10 m long and 0.43 m width. The mowed sample was weighed, then oven dried (65 °C during 72 h) to determine the dry matter (DM) content. Samples randomly hand picked up on the pastures were analysed to determine their nutritional and mineral compositions.

### 2.2.4. Statistical Analysis

- Statistical Analysis In Vitro

The statistical analyses were made using GraphPad Prism 6 (GraphPad Software, La Jolla, CA, USA). The data were analysed according to two-way ANOVA. The following model was applied:

$$Y_{ij} = Gr_i + T_j + Gr_i \times T_j + e_{ij}$$

where the effects were Gr = group effect (i = 1 and 2: control vs. MIX 3.0) and T = time of fermentation (j = 6 h and 24 h). The interactions between time and group were analysed, and $e_{ij}$ represents the residual error.

- Statistical Analysis In Vivo

The statistical analyses were performed using SAS 9.3 (SAS Institute Inc., Cary, NC, USA). The data were analysed according to the PROC MIXED procedure with repeated measures on random factor = animal and covariance analysis type compound symmetry.

The following model was applied.

$$Yijkl = \mu + Gr_i + S_j + LN_k + Cons_l + Gr_i \times S_j + e_{ijkl}$$

where $\mu$ = mean, Gr = group effect (i = 1 and 2: control vs. GQY), S = sampling (j from 1 to 3), LN = lactation number (k = 1 to 3 with 1 = primiparous, 2: $2^d$ lactation and 3 = more than 2 lactations) and the consumption of concentrate received at the ACS ($Cons_l$; 1 to 3) with 1 = cons < 1 kg/cow/d; 2 = cons from 1 to 2 kg/cow/d and 3 = cons > 2 kg/cow/d. The interactions between S and Gr were analysed, and $e_{ijkl}$ represents the residual error.

Yijkl was tested for methane predicted on the basis of mid infrared (MIR) spectra (g/cow/d and g/kg of milk), MY (kg/cow/d), F and P % and milk urea (mg/L). The test ANOVA1 was used for the analysis of the difference of methane emissions in breath samples analysed by the Guardian.

The statistical significance level was set at $p < 0.05$, *p*-value $p > 0.05$ and <0.10 was considered as trend.

## 3. Results

### 3.1. In Vitro

The parameters measured during the incubation of ruminal fluid showed significant differences between MIX 3.0 and the control (Table 1). Digestibility of NDF and ADF after 24 h was increased by MIX 3.0 addition by 5% and 4.4%, respectively in comparison with the control. The number of protozoa decreased from $3.11 \times 10^5$/mL (control) to $1.96 \times 10^5$/mL (MIX 3.0) after 6 h incubation. After 24 h, this decrease was even more marked as protozoa number was almost halved ($4.09 \times 10^5$/mL, control, to $2.00 \times 10^5$/mL, MIX 3.0). Gas production was reduced by 2 for methane (from 22.8 mL/g, control, to 14.4 mL/g, MIX 3.0) and by 59% and 53% in MIX 3.0 compared with the control for $NH_3$ after 6 h and 24 h, respectively (Table 1). The production of propionic acid increased from 6 h to 24 h, but the difference observed between groups (+14.5% in MIX 3.0 after 6 h) tended to lessen with time (+7.2% in MIX 3.0 after 24 h) (Figure 1). The production of acetic and butyric acids were numerically decreased by the addition of MIX 3.0, but this difference did not reach the statistically significant level ($p < 0.05$). However, in the ruminal fluid incubated with MIX 3.0, the ratio of acetate/propionate was significantly decreased by 6% ($p < 0.001$) after 24 h (Table 2).

**Table 1.** Counting of protozoa, methane production ($CH_4$), fibre digestibility and ammonia ($NH_3$) concentration during in vitro fermentation.

|  | **Control** | **MIX 3.0** | **Statistical Significance** |
|---|---|---|---|
| dNDF (%) after 24 h | $45.2 \pm 3.3$ | $50.2 \pm 1.6$ | * |
| dADF(%) after 24 h | $38.7 \pm 2.1$ | $43.1 \pm 3.1$ | * |
| Protozoa ($\times 10^5$/mL) after 6 h | $3.11 \pm 2.1$ | $1.96 \pm 1.3$ | ** |
| Protozoa ($\times 10^5$/mL) after 24 h | $4.09 \pm 2.0$ | $2.00 \pm 1.9$ | *** |
| $CH_4$ (mL/g DM) | $22.8 \pm 2.1$ | $14.4 \pm 1.2$ | *** |
| $NH_3$ (mMol/L) after 6 h | $7.81 \pm 0.29$ | $4.58 \pm 1.94.0$ | *** |
| $NH_3$ (mMol/L) after 24 h | $14.86 \pm 2.81$ | $7.81 \pm 0.74$ | *** |

Abbreviations: dNDF, digestible neutral detergent fibre; dADF, digestible acid detergent fibre; DM, dry matter; $CH_4$, methane; $NH_3$, ammoniac; ns, not significant. *: $p < 0.05$; **: $p < 0.01$; ***: $p < 0.001$.

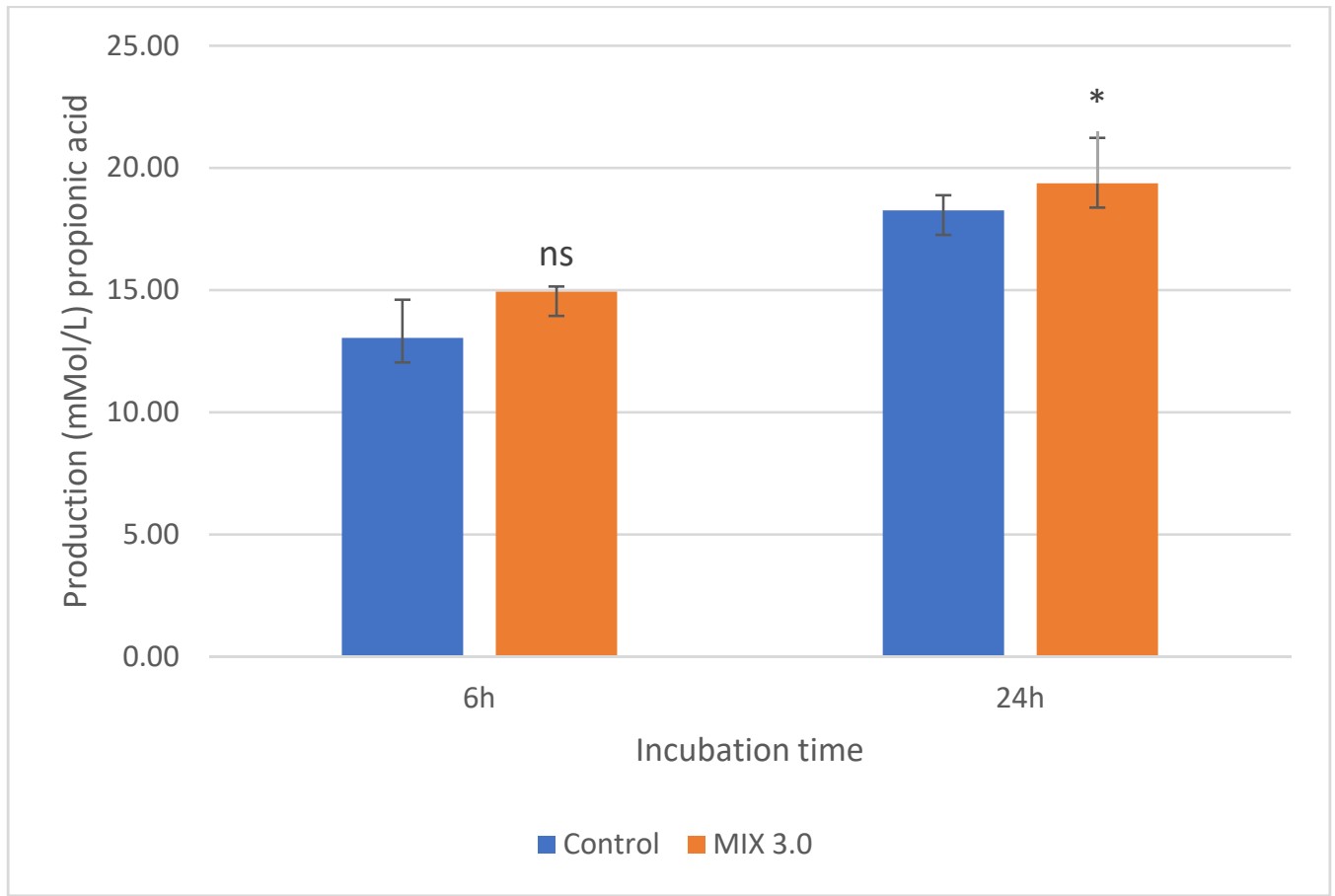

**Figure 1.** Comparison of production of propionic acid after 6 h and 24 h of incubation of ruminal fluid with and without MIX 3.0. Abbreviations: ns: not significant; *: significant at *p*-value < 0.05.

**Table 2.** Production of acetic, propionic and butyric acid after 6 h and 24 h of incubation of ruminal fluid with and without MIX3.0.

|  | Control | MIX 3.0 | Statistical Significance |
|---|---|---|---|
| Acetic acid (mMol/L) after 6 h | 35.26 ± 4.5 | 34.18 ± 3.9 | ns |
| Acetic acid (mMol/L) after 24 h | 50.16 ± 2.6 | 49.54 ± 10.0 | ns |
| Propionic acid (mMol/L) after 6 h | 13.04 ± 1.6 | 14.94 ± 0.2 | ns |
| Propionic acid (mMol/L) after 24 h | 18.26 ± 0.6 | 19.57 ± 1.9 | * |
| Butyric acid (mMol/L) after 6 h | 11.26 ± 1.9 | 9.56 ± 1.1 | ns |
| Butyric acid (mMol/L) after 24 h | 14.19 ± 0.7 | 13.00 ± 3.8 | ns |

ns: not significant. *: $p < 0.051$.

### 3.2. In Vivo

#### 3.2.1. Grazing

The average stay on pasture was 4.8 ± 1.5 days. The average grass height was 6.2 ± 1.9 cm (min: 3.0 cm, Max: 12.3 cm). The average nutritional value of grass is presented in Table 3 High energy and protein content were noted (VEM: 1021 ± 34 g/kg DM, CP: 26 ± 30 g/kg DM). Grass availability was estimated at 11.5 and 9.6 kg DM/cow/d in August and September, respectively. In complement, cows received on average 11.2 kg DM cereal crop silage. The daily total diet thus reached 21.7 kg DM forages on average over the whole trial period. The nutritional values and mineral contents of the 3 components of cows' diet are provided in Tables 3 and 4.

### 3.2.2. Mineral Inputs

On average, 129 ± 79 g/day of QY was consumed per cow. The most important consumption was observed in the three first days of the trial (396 g/day). After this transition period, the average QY intake reached 118 ± 52 g/day and was still very variable. Factors leading to these variations could not be identified. The nutritional inputs of trace elements were estimated (Table 5) on the basis of the average consumption of QY and concentrate (1.20 kg DM/cow/d) and were bcompared to nutritional recommendations edited by NRC (2001) [11], updated in 2019 [26] and Meschy, 2007 [27]. As some discrepancies were noticed between these two sources, the ratio inputs/requirements were calculated for both. Large amounts of Se (1.18 mg/d) and Co (4.72 mg/d) are provided by QY, representing 49% of requirements and 85% for Se and Co, respectively, compared to the most severe reference [26]. We must underline that the inputs in Se from QY intake represent 36% of the inputs of the total diet. The comparison between the diets fed to each group, taking into account the different consumption in concentrate, is provided in Table 6.

**Table 3.** Nutritional values of the cereal crop silage, concentrate and grazed grass.

| g/kg DM | Grazed Grass August | Grazed Grass September | Cereal Crop Silage | Concentrate |
|---|---|---|---|---|
| DM (%) | 21.7 ± 4.3 | 16.9 ± 2.9 | 32 | 88.5 |
| CP | 233 ± 29 | 260 ± 25 | 102 | 200 |
| Cellulose | 196 ± 12 | 204 ± 21 | 297 | 115 |
| NDF | 420 ± 12 | 411 ± 29 | 519 | 278 |
| ADF | 256 ± 13 | 261 ± 22 | 341 | 133 |
| Lignin | 49 ± 4 | 48 ± 5 | | |
| VEM | 1029 ± 30 | 1013 ± 30 | 760 | 870 |
| Total Ashes | 110 ± 6 | 142 ± 28 | 58 | 106 |

Abbreviations: DM, dry matter; CP, crude protein; NDF, neutral detergent fibre; ADF, acid detergent fibre; VEM, Voeder Eenheid voor Melk: Dutch unit representing the Net energy for lactation: 1000 VEM = 6.9 MJ NEL.

**Table 4.** Mineral content of the different feedstuffs from cows' diet.

| Content (mg/kg DM) | Grazed Grass | Cereal Crop Silage | Concentrate |
|---|---|---|---|
| Zinc | 27 | 61 | 115 |
| Manganese | 20.5 | 25 | 83.3 |
| Copper | 9.4 | 5.6 | 23.1 |
| Cobalt | 0.1 | 0.03 | 1 |
| Selenium | 0.1 | 0.05 | 0.4 |
| Iodine | 0.1 | 0.3 | 1.6 |

**Table 5.** Daily trace mineral inputs on average consumption of QY (mg/cow/d) and of concentrate (g/cow/d) compared to nutritional requirements edited by NRC (2001), updated in 2019 [26] and Meschy (2007) [27].

| | Inputs by Concentrate Intake 1.20 kg DM | Inputs by Grazed Grass Intake 10.5 kg DM | Inputs by Cereal Crop Intake 11.2 kg DM | Total (mg/d) | Inputs by QY Intake (mg/d) | Total (mg/d) GQY | Requirements (mg/d) [26] | Requirements (mg/d) [27] |
|---|---|---|---|---|---|---|---|---|
| Zinc | 137 | 284 | 683 | 1105 | 236 | 1133 | 990 | 1195 |
| Manganese | 100 | 215 | 280 | 595 | 28 | 831 | 582 | 1195 |
| Copper | 28 | 99 | 63 | 189 | 59 | 248 | 506 | 445 |
| Cobalt | 1.20 | 1.05 | 0.34 | 2.59 | 4.72 | 7.31 | 8.80 | 6.60 |
| Selenium | 0.48 | 1.05 | 0.54 | 2.06 | 1.18 | 3.25 | 6.60 | 2.2 |
| Iodine | 1.92 | 1.05 | 3.36 | 6.33 | 11.8 | 18.13 | 9.92 | 12.0 |

Abbreviations: QY, Qualix Yellow; GQY, group Qualix Yellow.

### 3.2.3. Production Performance

The average live weight was similar in the groups during the trial (GQY: 656 ± 14 vs. GC: 644 ± 14 kg; ns). The interaction effect month X group was significant and showed a

gain in live weight from August to October in each group. However, this observation was not confirmed by BCS values, which stayed stable over time in both groups. No difference in BCS between groups was noted. Milk yield tended to increase in GQY (MY GQY: 24.22 ± 1.02 vs. GC: 21.82 ± 1.00 kg/cow/d), while concentrate consumption dropped (Concentrate consumption GQY: 1.20 ± 0.03 kg DM/cow/day vs. GC: 1.49 ± 0.03 kg DM/cow/day; $p < 0.001$) (Table 7). Neither milk composition nor ECM production were altered by the treatment. Milk urea was more elevated in GQY compared with GC (GQY: 383 ± 9 mg/L vs. GC: 356 ± 9 mg/L; $p < 0.05$).

**Table 6.** Total nutritional inputs from the diets fed to each group, i.e., GQY and GC based including 10.5 kg DM grazed grass and 11.2 kg cereal crop silage. Concentrate complementation: 1.2 kg DM for GQY and 1.49 kg DM for GC.

| | Total Diet Fed to GQY | Total Diet Fed to GC |
|---|---|---|
| Total kg DM fed per day | 23 | 23.2 |
| Nutritional inputs g/kg DM | | |
| DM | 290 | 300 |
| CP | 173 | 174 |
| cellulose | 242 | 241 |
| NDF | 457 | 456 |
| ADF | 291 | 290 |
| VEM | 882 | 885 |
| Mineral inputs mg/kg DM | | |
| Zinc | 49 | 49 |
| Manganese | 36.1 | 26.7 |
| Copper | 10.8 | 8.4 |
| Cobalt | 0.32 | 0.12 |
| Selenium | 0.14 | 0.09 |
| Iodine | 0.79 | 0.29 |

Abbreviations: GC:, group control; GQY, group Qualix Yellow; DM, dry matter; CP, crude protein; NDF, neutral detergent fibre; ADF, acid detergent fibre; VEM:,Voeder Eenheid voor Melk: Dutch unit representing the Net energy for lactation: 1000 VEM = 6.9 MJ NEL.

**Table 7.** Milk yield and concentrate consumption in both groups. Milk composition and methane emissions are also reported. The upper section shows results from the analysis of daily individual values, while the lower section shows results from monthly milk quality analysis. Values are LS means ± SE.

| | Group | | Statistical Significance | | |
|---|---|---|---|---|---|
| | **GQY** | **GC** | **Gr Effect** | **S Effect** | **Gr X S** |
| MY (kg/cow/d) | 24.22 ± 1.02 | 21.82 ± 1.00 | $p < 0.1$ | *** | *** |
| Concentrate consumption (kg DM/cow/d) | 1.20 ± 0.03 | 1.49 ± 0.03 | *** | *** | *** |
| ECM (kg/cow/d) | 25.28 ± 0.92 | 24.12 ± 0.98 | ns | *** | *** |
| F% | 4.04 ± 0.10 | 4.25 ± 0.10 | ns | *** | ns |
| P% | 3.43 ± 0.07 | 3.46 ± 0.07 | ns | *** | ns |
| Urea (mg/L) | 383 ± 9 | 356 ± 9 | * | *** | *** |
| Methane (g/cow/day) | 444 ± 13 | 445 ± 13 | ns | *** | ns |
| Methane (g)/kg of milk | 17.98 ± 0.97 | 20.25 ± 0.94 | $p < 0.1$ | *** | *** |
| Methane (g)/kg of ECM | 17.73 ± 0.89 | 19.26 ± 0.87 | ns | *** | *** |
| Methane in breath samples per visit to ACS (ppm) | 0.100 ± 0.016 | 0.112 ± 0.013 | *** | ns | ns |

Abbreviations: MY, milk yield; Gr, Group; S, sampling; ECM, energy corrected milk; F, fat; P, protein; ns, not significant. *: $p < 0.05$; ***: $p < 0.001$.

Although predicted methane emissions per cow (g/d) showed no significant difference between groups, methane emissions/kg of milk tended to decrease in GQY (GQY: $17.98 \pm 0.97$ g methane/kg of milk vs. GC: $20.25 \pm 0.94$ g methane/kg of milk; $p < 0.1$). A declining trend was observed in breath methane emissions ($0.100 \pm 0.016$/visit to the ACS in GQY vs. $0.112 \pm 0.013$ ppm/visit in GC, $p < 0.1$).

Table 7 shows that month effect was significant on several parameters. Figure 2 shows that the milk yield of GQY stayed more stable with a decrease from $25.37 \pm 1.02$ in August to $23.09 \pm 1.04$ kg/cow/d in October, i.e., 9% decrease. On the contrary, a decrease by 16% from August to October was recorded in GC (August: $23.41 \pm 1.10$ vs. October: $19.54 \pm 1.0$ kg/cow/day).

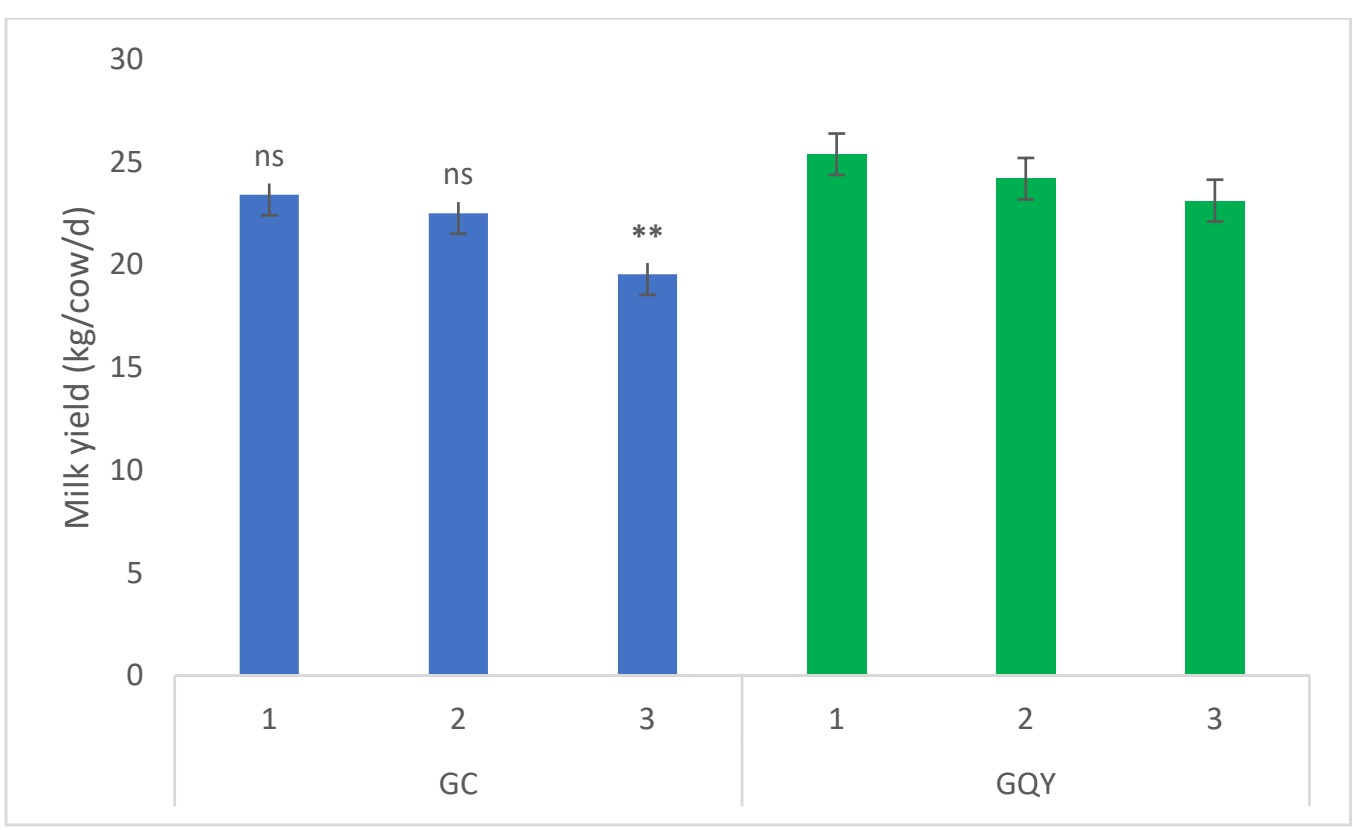

**Figure 2.** Evolution of milk yield over the 3 samplings in the group receiving the QY (GQY) and the group control (GC). Abbreviations: ns, not significant; **: $p < 0.01$.

Variations in MY induced a sampling effect and a sampling X group effects for methane emissions/kg of milk and per kg of ECM with a significant difference observed in S3 (Methane/kg of milk GQY: $18.6 \pm 1.1$ g/kg of milk vs. GC: $23.2 \pm 1.1$ g/kg of milk, $p < 0.01$; Methane/kg of ECM GQY: $17.7 \pm 0.9$ g/kg of ECM vs. GC: $21.1 \pm 0.9$ g/kg of ECM; $p < 0.001$). Fat and protein levels increased at the same rate in each group: F% from $3.92 \pm 0.9\%$ from S1 to $4.34 \pm 0.9\%$ in S3, P% from $3.27 \pm 0.5\%$ from S1 to $3.66 \pm 0.5\%$ for S3.

Average urea level was higher in S2 and then decreased from $394 \pm 9$ mg/L to $374 \pm 9$ mg/L in S3. This was mainly due to a sharp increase in GQY compared with GC (GQY: $319 \pm 13$ mg/L in S1 to $402 \pm 12$ mg/L in S3; GC: $361 \pm 13$ mg/L in S1 to $346 \pm 12$ mg/L in S3).

Mean residues from the faecal samples represented 4.7% and 4.6% of the total sample weight in GQY and GC, respectively.

## 4. Discussion

This paper aimed to evaluate the effects of the compound QY. The goal of this compound is to optimize the inputs in mineral and trace elements and to improve ruminal function. It is composed of MIX 3.0 up to 20% and of several mineral and trace elements. The QY is commercialized as a licking bucket. In a first step, in vitro experimentations were conducted to assess the effects of MIX 3.0 on ruminal fermentations. The dosage of MIX 3.0 to be tested was determined by preliminary research, including evaluation of the palatability. As the in vitro results were promising, in vivo tests were led with several objectives: The first was to verify that the amounts consumed by the cows provided efficient complementation of their diet in trace and mineral elements. The second objective was to confirm the effects on ruminal fermentation linked to the component MIX 3.0 included in QY. Combining the results of in vitro and in vivo trials is essential to give a complete overview of the potential interests of this compound in its final formulation, i.e., licking bucket. In vitro trials presented the conclusion that the addition of MIX 3.0 modified the ruminal fermentation processes, with increased production of propionate and decreased ration acetate/propionate. This change in fermentation pattern is one of the possible means to mitigate ruminal methane production [3,28,29]. Increased digestibility of ADF and NDF indicated that cellulolytic flora was more efficient with the use of MIX 3.0. In parallel, methane and $NH_3$ production were reduced during in vitro fermentation with MIX 3.0. These effects comply with the literature. The effects of yeasts on ruminal fermentation have been confirmed by other studies [3,28,29], which also established an increased fibre digestion. Certain plants added to MIX 3.0 are likely to modify ruminal flora, and thus the fermentation pattern. For example, *Thymus vulgaris* L. and *Allium sativum* L. have demonstrated strong antimicrobial properties [30–33] that induce a shift in ruminal flora and consequently in the ruminal fermentation pattern. Moreover, goosefoot (*Chenopodium quinoa* W.) is recognized for its high content in saponins [34]. This specificity could explain the defaunation of protozoa observed during ruminal fluid incubation [35].

This marked decline in protozoa counts ($-50\%$) is regularly cited by the literature as a means to decrease methane emissions by lowering the transfer of hydrogen from protozoa to *Archea* spp., enabling methane production [36,37]. Another explanation for the drop of methane production observed could be the enhanced production of propionate. This metabolic pathway consumes hydrogen issued from ruminal fermentation, making it less available for methanogenesis [38,39]. Methane production generates losses in energy that can be estimated at 2 to 12% of dietary gross energy [37,40,41] and consequently could lead to a decrease in feed conversion. A decrease in methane emissions per kg of consumed concentrate may indicate potential improvement of this parameter. Yeast and plant extract have presumably combined their effects to optimize the ruminal function.

The in vitro trials highlighted a sharp decrease in $NH_3$ production after 6 h and 24 h, respectively. The measurement of $NH_3$ concentration in ruminal fluid is an indicator of protein efficiency [42]. Lowered $NH_3$ production is linked to a decrease in intra-ruminal deamination and thus to an increase in undegradable ruminal proteins [30,43]. The strong deamination power of some plants included in the MIX 3.0, e.g., *Thymus vulgaris* L. [32,44] could explain this effect. Another explanation could be the antimicrobial effect of *Chenopodium quinoa* W. on protozoa and on proteolytic bacteria [45]. Conversely, ruminal microorganisms require ruminal $NH_3$ for the growth and synthesis of microbial proteins [38,42]. Despite the decrease in $NH_3$ observed with MIX 3.0 after 6 h, the measured level remains within the range of values of $NH_3$ concentration (between 3.5 mM and 6 mM) necessary for production of microbial protein [42].

Grazing cows were targeted by the in vivo trial. Grazing is very common in Western Europe; it decreases feeding costs and offers ecosystem services [46]. The animals also received concentrate and cereal crop silage to complete the diets to achieve the expected production performance. On the basis of the in vitro results, decreased methane emissions, a rise in MY and decreased milk urea through the improvement of feed conversion efficiency in GQY were expected. In fact, this group demonstrated a trend in increased milk yield

by 2.4 kg/cow/d, while concentrate consumption was lower (−16%). Moreover, the milk yield remained more stable over the trial period than those of GC. Despite stable MY and decreased concentrate consumption, no difference in live weight was observed between GQY and GC. All these results lead us to assume that feed efficiency was improved with QY supplementation in accordance with in the vitro results. A declining trend in methane emissions (g/kg of milk or/kg of ECM) and a significant drop in methane (ppm) in breath samples in GQY were observed. The differences in methane/kg milk and methane/kg ECM were even more pronounced in October. This leads us to presume that an adaptation period is necessary for the compound to take effect.

Nitrogen efficiency does not appear to be improved in view of the higher average milk urea concentrations in GQY, although there is lower concentrate consumption and higher MY. It is noteworthy that inputs in the components of MIX 3.0, i.e., plants and yeast extracts were less abundant in the in vivo trials as MIX 3.0 represents 20% of the total composition of QY.

Mean residues from the faecal samples demonstrated no difference between the groups.

According to the literature [25,47,48], digestive efficiency can be considered very good. The consumption of QY complied with the recommendations of the company (118 g/cow/d). The appetence of the product was satisfactory as the intake was fairly high in the first three days of the trials. Nevertheless, the ingestion was very variable, and no explanation can be provided through investigation of several parameters. Additional mineral inputs reached through the allowance of QY allowed an increase of 36% (Se) and 64% (Co) of nutritional mineral inputs in GQY compared with GC, yet most permanent grasslands are deficient in trace elements, i.e., Zn, Cu, Mn and Se [26,49], making sup-plementation at grazing with QY valuable. The amounts of QY consumed by the cows helped to increase the mineral inputs of the diet. Depending on the reference taken into consideration, some adjustments could be still necessary. The most important of these is the copper intake. However, copper supplementation has narrow safety margins [26] which have led to caution about increasing levels in a feed additive. Needs in selenium were very differently estimated following the references, so that the supplementation complied with the recommendations of Meschy [27] but was still insufficient following NRC updates [26]. The effects of a longer administration of QY should confirm these preliminary observations, in terms of both ruminal function and mineral intake.

## 5. Conclusions

The results of in vitro tests were encouraging, demonstrating a huge decrease of methane and $NH_3$ emissions and an increase in propionate production, with the use of MIX 3.0 being part of the composition of QY. These outcomes lead us to expect a large impact during in vivo implementation. It was lower than expected, although the studied compound met most of its objectives. Milk yield (kg/cow/d) was enhanced and more persistent, while methane emissions (g/kg milk–g/kg ECM) decreased in the supplemented group. Nevertheless, this trend was observed after several weeks of use. These preliminary results should be confirmed by trials held over a longer period.

## 6. Patents

MIX 3.0 is under patent EP3558027.

**Author Contributions:** Conceptualization, A.-L.L.K. and I.D.; methodology, A.-L.L.K. and I.D.; vali-dation, I.D., S.P. and F.L.; resources, I.D. and A.-L.L.K.; data curation, F.L. and S.P.; writing—original draft preparation, F.L.; writing—review and editing, F.L.; visualization, F.L. and S.P.; supervision, I.D. and A.-L.L.K.; project administration, I.D.; funding acquisition, I.D. All authors have read and agreed to the published version of the manuscript.

**Funding:** This research was partially funded for in vivo trials by the Centre Mondial de l'Innovation, Groupe Roullier. This funding includes limited contribution to feeding costs.

**Institutional Review Board Statement:** The study was conducted according to the national guidelines relative to animal welfare.

**Informed Consent Statement:** Not applicable.

**Data Availability Statement:** The datasets generated for this study are available on request to the corresponding author.

**Conflicts of Interest:** Anca-Lucia Laza Knoerr is an employee of the company Centre Mondial de l'Innovation—Groupe Roullier. She was responsible about the conceptualization, methodology, validation, resources and supervision of in vitro trials; Sandra Point is an employee of the company Centre Mondial de l'Innovation—Groupe Roullier. She was responsible for validation, data curation and visualization of in vitro trials. MIX 3.0 is a formulation developed by Roullier.

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
