# Peer review of "In Vitro and In Vivo Evaluation of the Effects of a Compound Based on Plants, Yeast and Trace Elements on the Ruminal Function of Dairy Cows"

_2624-862X, doi:10.3390/dairy2040043_

Round 1

Reviewer 1 Report

The authors conducted a good research:

Please add the location of the current research (latitude: E' ; N') climate , where this research was conducted.
VFA - for the future, with Agilent's GC, it is worth introducing VFA with GC. A determination of NH3 with an ionometer. In such a case, the work could be "sold" for a larger IF 

Overall, the experiments are well designed and executed with data analyzed by appropriate methods.

Reviewer 2 Report

This paper evaluates in vitro and in vivo two commercial compounds for dairy cows. The manuscript is a bit confusing because it is not clear whether the compound QY or MIX 3.0 or both are being evaluated, whether MIX 3.0 is an additional ingredient for QY or is a primary ingredient. Some effects, could be attributable to other ingredients of QY? The Material and Methods section needs a considerable attention then, results would be more understandable. The Discussion lacks new insights and merely ‘confirms’ and ‘describes’ results relative to earlier studies.

Specific comments

L50: The authors need to define QY the first time and not in the second time in L55

L51-53: In binomial nomenclature, the standard designating the scientist who first published the name must be included, as the authors did with Inula helenium L. As well as, the specific name must be written in lower case: Inula helenium L., Artemisia absinthium L. (and not Arthemisia Absynthum)

L55: Qualix Tellow must be defined before (see L50)

L66: How many cows were used as donors? Each one of them were used as an experimental unit? Or ruminal fluids of donor cows were mixed?

L68: The proportion of ingredients are in dry matter basis or fresh matter basis?

L73: The reference of Menke and Steingass (1988) is not in reference list

L74: Again, ingredients are in dry matter basis or fresh matter basis? Therefore, the ideal approach in an in vitro assay is that the diet of the donor cows and the test flasks was the same. Small differences can be seen.

L77: Delete one dot.

L78: The 4 repetitions are lab replicates or animal replicates? If they are lab replicates, the animal variability is lost.

L79: References 13 and 14 are the same (and the same as 12).

L80: How long gas production was measured? Are the authors sure that Ankom system is the best alternative for gas measurement?

L85: Which other volatile bases were determined? The results are not show. If there are not results of them, then don't make mention their analysis.

L94: Check the reference: 16? If yes, delete reference 15 and renumber the remaining references.

L95 and on: Check headings and subheadings numeration.

L99: How long was the adaptation period?

L117: The herbage disappearance technique is only suitable for estimating forage intake for groups of animals on well-managed, rotationally stocked pastures, with short grazing periods. Estimates by the herbage disappearance method requires frequent sampling to avoid the grass regrowth. However, estimation of forage intake based on energy requirements for animal performance is useful when evaluating intake of lactating dairy cows grazing pastures and substantial fieldwork is not required. The authors shown in their work that they have estimated the energy requirements of cows and the energy value of feeds. Then, they could estimate the grazed grass intake with a more accurate way. This reviewer anticipates that, if sampling frequency is high and energy estimations are accurate, both methods are suitable.

L120: Nutritional values of grazed grass, concentrate, and cereal crop silage (oats, triticale, peas)

L134: Two methods of methane emissions measurements are described. But seems that only results of a method are shown. Which method? Please, clarify. Again, if there are not results of one of them, then don't describe it.

L139: Seven days seem a very short period to have accurate measurement of methane emissions.

L140-143: If animal groups were physically separated (L105), how is it possible take measurements of methane ‘in the same automatic concentrate supplier’? But after the authors say that ‘devices were changed from one station to another’. Please, clarify that.

L166-167: Binomial names in italic and including the initial of first publisher name.

L188: 1 and 2

L188: Just to be clear, S=sampling (1 to 3) means month of sampling? Three samples? But the assay lasted two months!! How can that be? (See below).

L205 and 207: The authors say: ‘The number of protozoa was […] (control) to 1.96x105/ml (QY) […] control to 2.00x105/ml – MIX 3.0)’. Then, what is being evaluated? QY or MIX 3.0? (See general comment).

L210: Only propionic acid results are shown. But acetic and butyric acids are important in a study about methanogenesis. Please, include them. Branched volatile fatty acids could also be interesting to understanding some results.

Table 2. Significance column with p value or *, delete one of them. In number of protozoa rows, write 5 as superscript. In methane and ammonia rows, write subscripts. In footnote, ammonia.

L234: Provide DM intake of concentrate

L236 and Table 3: Trace mineral intake is the basis of this work. However, mineral analysis is not described in Materials and methods section. Please, provide it. A table with the mineral composition of each feed is also necessary.

L241: the average consumption of grass, cereal crop silage, QY, and concentrate

L242-243 and table 3: I am not sure about the exact rules for this Journal but references should be written only in square brackets. Please check this.

Table 3: Please provide DM intake of concentrate. If grass availability was estimated at 7.5 and 6.5 kg DM/day (L233), why the mineral inputs by grass intake were estimated on basis to 11.5 kg DM?

L252: The study lasted 66 days (two months and six days; L99), if milk was sampled once a month (L125), only two milk samples were taken (considering at least two weeks of adaptation period). But results of three samplings is shown. Please, clarify this point.

L263: The methane emissions per cow results, what methodology was? Were the results of both methodologies comparable with each other?

L268: The Figure 2 shows…

Figure 2: Three samplings? Once monthly? The study lasted only two months. This is a very important issue which must be clarified.

L301-304: The authors say: ‘The effect of MIX 3.0 was studied on ruminal fermentation in vitro’ and ‘the effects of the administration of QY on production performance of grazing dairy cows’. Then, the effects studied were of MIX 3.0 or QY? The other components of QY did not effect? This is difficult to understand and believe.

L321: What about acetate and butyrate production? They are of great importance on methane production.

L327: A reference is necessary for this statement.

L333: Binomial names in italic

L336: MIX 3.0

L341: The animals received also concentrate and cereal crop silage to complete the diets

L343: The authors say: ‘risen MY and decreased milk urea through the improvement of feed conversion efficiency in GQY’. According table 4 and L231, milk urea was more elevated in GQY than GC. Besides, the high urea values in both groups clearly indicate a low protein efficiency.

L351: The authors say: ‘The difference [methane (ppm) in breath samples] was even more pronounced in October’. How is that possible if ‘the measurements were performed from 19 to 25 September’ (L139)?

L413: 2001

L414-422: References 12, 13, and 14 are the same. Check

L423: I think that reference 15 is not correct. It is not match with the text.
